A systematic analysis of anti-diabetic medicinal plants from cells to clinical trials

Omale Simeon 1 2
http://orcid.org/0000-0002-4630-0442 Amagon Kennedy I. 2
Johnson Titilayo O. 3
Bremner Shaun Kennedy 4
http://orcid.org/0000-0001-6571-2875 Gould Gwyn W. 4 gwyn.gould@strath.ac.uk
1 African Centre for Excellence in Phytomedicine, University of Jos , Jos , Nigeria
2 Department of Pharmacology and Toxicology, Faculty of Pharmaceutical Sciences, University of Jos , Jos , Nigeria
3 Department of Biochemistry, Faculty of Basic Medical Sciences, University of Jos , Jos , Nigeria
4 Strathclyde Institute of Pharmacy and Biomedical Sciences, University of Strathclyde , Glasgow , United Kingdom
Kuddus Mohammed
Electronic publication date: 2023 Jan 5
Publication date: 2023
Volume: 11
Electronic Location ID: e14639
Received 2022 Aug 15; Accepted 2022 Dec 5
Copyright: © 2023 Omale et al.
Copyright year: 2023
Copyright holder: Omale et al.
License: This is an open access article distributed under the terms of the Creative Commons Attribution License, which permits unrestricted use, distribution, reproduction and adaptation in any medium and for any purpose provided that it is properly attributed. For attribution, the original author(s), title, publication source (PeerJ) and either DOI or URL of the article must be cited.
License URL: https://creativecommons.org/licenses/by/4.0/

Keywords: Medicinal plants, Insulin action, Diabetes, Glucose transport, Akt, Insulin signaling, Phytochemicals

Funding: African Research Excellence Fund AREF-308-OMALE-F-C0818 Diabetes UK 18/0005847 This work was supported by a grant from the African Research Excellence Fund to Simeon Omale and Gwyn Gould (AREF-308-OMALE-F-C0818) and a grant from Diabetes UK (to Gwyn W Gould; 18/0005847). The funders had no role in study design, data collection and analysis, decision to publish, or preparation of the manuscript.

==============================
Background

Diabetes is one of the fastest-growing health emergencies of the 21st century, placing a severe economic burden on many countries. Current management approaches have improved diabetic care, but several limitations still exist, such as decreased efficacy, adverse effects, and the high cost of treatment, particularly for developing nations. There is, therefore, a need for more cost-effective therapies for diabetes management. The evidence-based application of phytochemicals from plants in the management of diseases is gaining traction.

Methodology

Various plants and plant parts have been investigated as antidiabetic agents. This review sought to collate and discuss published data on the cellular and molecular effects of medicinal plants and phytochemicals on insulin signaling pathways to better understand the current trend in using plant products in the management of diabetes. Furthermore, we explored available information on medicinal plants that consistently produced hypoglycemic effects from isolated cells to animal studies and clinical trials.

Results

There is substantial literature describing the effects of a range of plant extracts on insulin action and insulin signaling, revealing a depth in knowledge of molecular detail. Our exploration also reveals effective antidiabetic actions in animal studies, and clear translational potential evidenced by clinical trials.

Conclusion

We suggest that this area of research should be further exploited in the search for novel therapeutics for diabetes.

Introduction

Statement of the problem

Diabetes mellitus is a metabolic disorder characterized by sustained hyperglycemia with numerous macrovascular and microvascular complications (Kharroubi, 2015; Stadlbauer et al., 2021; Zheng, Ley & Hu, 2018). Depending on the etiology, diabetes has been classified broadly into type 1 diabetes mellitus (T1DM), resulting from a deficiency in insulin production, and type 2 diabetes mellitus (T2DM), a defect in insulin action (Kharroubi, 2015). T1DM accounts for around 10% of cases, while T2DM accounts for about 90%. T2DM and its complications have contributed to a significant decrease in life expectancy (Zheng, Ley & Hu, 2018; Ogurtsova et al., 2017). The latest data suggests that 1 in 10 adults are living with diabetes, of which almost half are undiagnosed (Zheng, Ley & Hu, 2018), representing around 537 million citizens, and diabetes contributes to 1 in 9 deaths (Stadlbauer et al., 2021). These figures are projected to continue to rise. It is estimated that most countries devote 5–20% of healthcare expenditure to diabetes (Lin et al., 2020). The global spending to treat diabetes and its complications was US$760 billion in 2019, projected to increase to US$825 billion by 2030 (Stadlbauer et al., 2021; Modi, 2007). The disturbing increase in the prevalence of diabetes is a call for an augmented approach to the management of T2DM (Stadlbauer et al., 2021; Williams et al., 2020).

The conventional management of diabetes involves lifestyle modifications to control contributing factors such as obesity, hypertension and hyperlipidemia, based on the patient’s awareness (Pernicova & Korbonits, 2014; Proks et al., 2018; Farzaei et al., 2017; Soccio, Chen & Lazar, 2014), and the use of hypoglycemic agents by healthcare providers (Hedrington & Davis, 2019; Mogensen, 2007; Hinnen et al., 2006). Approaches to using antihyperglycemic agents to control hyperglycemia in diabetic patients involve different targets. For example, the sulfonylureas (e.g., chlorpropamide) and newer secretagogues (e.g., glipizide) increase insulin output by blocking the K+-ATPase channel of the pancreatic β-cell (Hedrington & Davis, 2019; Wajcberg & Tavaria, 2009; Bailey, 2015). The biguanides (e.g., metformin) act through inhibition of hepatic gluconeogenesis and promoting glycogenesis with increased insulin sensitivity (Hedrington & Davis, 2019; Hinnen et al., 2006; Bailey, 2015). Insulin sensitizers (e.g., thiazolidinediones) potentiate insulin action on muscle, adipocytes, liver and other tissues by selectively binding to peroxisome proliferator-activated receptor gamma (PPARγ) (Hedrington & Davis, 2019; Triggle & Ding, 2014). Others include α-glucosidase inhibitors (e.g., acarbose), which competitively inhibits intestinal α-glucosidase and pancreatic α-amylase with a resulting decrease in postprandial plasma glucose (Hedrington & Davis, 2019; Cetrone, Mele & Tricarico, 2014; Campbell, 2000). Incretin mimetics (e.g., exenatide) control postprandial insulin secretion by binding to the pancreatic glucagon-like peptides-1(GLP-1) receptors leading to increased glucose-dependent insulin secretion from the β-cells (Mogensen, 2007; Campbell, 2000; Harding et al., 2019). Exenatide also restores first-phase insulin secretion in patients with T2DM and promotes β-cells proliferation and islet neogenesis (Harding et al., 2019; Ogurtsova et al., 2017). The use of insulin for immediate glycemic control has been reserved for emergency situations (Mogensen, 2007; Bommer et al., 2017). Undoubtedly, these approaches have improved diabetic care over time, but several limitations still exist, such as decreased efficacy, adverse effects, and high cost of treatment (Michel, Abd Rani & Husain, 2020; Verma, 2014).

Medicinal plants as medicines in diabetes treatment

The global worsening of morbidity and mortality from diabetes (Zheng, Ley & Hu, 2018; Chaudhury et al., 2017; Modi, 2007; Md Sayem et al., 2018; Schreck & Melzig, 2021; Süntar, 2020) justifies the need for more diversified research for new therapies. Throughout human history, medicinal plants have been used for the prevention and treatment of both human and animal diseases (Balunas & Kinghorn, 2005; Majolo et al., 2019; Khazir et al., 2014; Ahmad et al., 2015). Medicinal plants have been recognized as a stable source for drug discovery since ancient times (World Health Organization, 2019; Singhal, Bangar & Naithani, 2012; Mathew & Subramanian, 2014; Martins & Brijesh, 2018) and The World Health Organization has reported an increased patronage of natural and medicinal plant drug products (World Health Organization, 2019). Many modern drugs are obtained from medicinal plants and further purified or optimized using structure-activity relationship-driven drug design and pharmacokinetic parameters (Singhal, Bangar & Naithani, 2012; Setorki, 2020; Sekhon-Loodu & Rupasinghe, 2019).

Evidence-based application of phytochemicals from plants in the management of diseases has received wide acceptability (Chukwuma et al., 2019; Mathew & Subramanian, 2014). For example, several reports of medicinal plants with anticancer activities have been published (Rabiei, Solati & Amini-Khoei, 2019; Sadino, 2018; Khan et al., 2012). Ethnopharmacological surveys of plants and phytochemicals with antihypertensive activities (Joseph & Jini, 2013; Shih, Lin & Lin, 2008; Al-Amin et al., 2006) have been well documented. There is also substantial literature of their utility in treatment of other chronic diseases such as Alzheimer’s (Son, Miura & Yagasaki, 2015; Morakinyo, Akindele & Ahmed, 2011), depressive disorders (Lai et al., 2015; Chien et al., 2009), Parkinson’s disease (Rabiei, Solati & Amini-Khoei, 2019) and diabetes (Bailey, 2015; Ogunbolude et al., 2009; Kanetkar, Singhal & Kamat, 2007).

Various plants and plant parts have been investigated for their hypoglycemic activities as potential medicine in the treatment of diabetes mellitus (Farzaei et al., 2017). By way of examples, phytocompounds from the fruit of Momordica charantia (bitter lemon) have been extensively studies for antidiabetic effects (Tiwari, Mishra & Sangwan, 2014; Kuroda et al., 2003; Shori, 2015). The roots of Zingiber officinale (ginger) exert antidiabetic and hypolipidemic effects on streptozotocin-induced diabetic rats (James, Stöckli & Birnbaum, 2021; Garvey et al., 1998; Ahmad, Choi & Lee, 2020). Bidens pilosa has been shown to reduce fasting blood glucose level and hemoglobin A1c (HbA1c) in clinical trials (Lai et al., 2015); three variants of B. pilosa were shown to possess anti-diabetic properties (Chien et al., 2009). The hydroethanolic extract of the seed of Parinari curatellifolia reduces plasma glucose levels and low-density lipoproteins in diabetic rats (Saini, 2010; Galochkina et al., 2019; Ogbonnia et al., 2009). The blood sugar reducing effects of Gymnema sylvestre popularly known as ‘gurmar’ (‘sugar destroyer’) has been widely studied (Kanetkar, Singhal & Kamat, 2007; Tiwari, Mishra & Sangwan, 2014). Phytochemical constituents of Glycyrriza uralensis (licorice) have been found to exhibit profound antidiabetic properties in experimental animals (Kuroda et al., 2003). While some studies do consider the potential molecular or cellular mechanisms of the antidiabetic effects (Ogurtsova et al., 2017; Vlavcheski et al., 2018), others focus on potential properties such as antioxidant (Ahmad, Choi & Lee, 2020; Galochkina et al., 2019) and anti-obesity (Kadan et al., 2018; Kamatou, Ssemakalu & Shai, 2021) effects without direct discussion of mechanism.

This review aims to collate, discuss, and present published data on the cellular and molecular effects of medicinal plants and phytochemicals on insulin signaling pathways to better understand the current trend in the use of plant products in the management of T2DM (Fig. 1). Furthermore, we have explored available information on the cell-biology of these medicinal plants that consistently produced hypoglycemic effects, with the intention of providing a reference point for the molecular basis of some of the more commonly used anti-diabetic plant extracts. We explored how these plant products might affect known insulin signaling systems and insulin effectors, and then extended our review into known effects on animal models and explored clinical trials of these compounds with the intention of providing a summary-view of related studies and a holistic overview of their use in rodent models or clinical trials. We conclude that plant products should be considered a vital tool in the armory for development of low-cost, effective anti-diabetic therapies.

Figure 1 Schematic representation of antidiabetic development from plant to clinical trial. The pathway from plant extract to effective therapy involves many steps. Plants, often identified from local knowledge/use, are a source of extract prepared using a range of approaches and the extracts screened using simple cell-based models such as Caco-2 cells or L6 skeletal myotubes for in vitro effects. Signaling pathways and effectors are used as surrogate assays for potential antidiabetic effects (e.g., glucose transport). Further work involves an examination of effects using rodent models of diabetes and clinical trials.

The pathway from plant extract to effective therapy involves many steps. Plants, often identified from local knowledge/use, are a source of extract prepared using a range of approaches and the extracts screened using simple cell-based models such as Caco-2 cells or L6 skeletal myotubes for in vitro effects. Signaling pathways and effectors are used as surrogate assays for potential antidiabetic effects (e.g., glucose transport). Further work involves an examination of effects using rodent models of diabetes and clinical trials.

Methodology

We used a range of search terms to scan Google Scholar, PubMed, Science Direct, NIH National Library of Medicine and Scopus to retrieve published literature on medicinal plants and phytochemical effects on insulin signaling and effector pathways. Search terms focused on known signaling systems involved in propagating insulin signals (e.g., proteinqrynm_32 kinase-B/Akt (hereafter referred to as Akt); phosphoinositide-3 kinase (PI3K); glycogen synthase kinase-3 (GSK-3); AMP-activated protein kinase (AMPK); protein tyrosine phosphatase 1 B (PTP1B)); known effector molecules or processes (e.g., glucose transporters (GLUT) and GLUT4 storage vesicles (GSVs; also known as GSC–GLUT4 storage compartment); glucokinase (GCK)); glucagon secretion; lipolysis; lipogenesis; hepatic glucose output; and other molecules implicated in insulin action of insulin sensitivity, such as peroxisome proliferator-activated receptor gamma (PPARγ). Searches were performed between December 2021 and April 2022. We excluded articles not in English and not freely available via our institution (in this case the University of Strathclyde; <0.2% of articles retrieved) and no time limitation for publication date was employed. Our searches aimed to capture papers which described a potential effect on either signaling systems (e.g., PI3K, Akt etc.) or a biological output (e.g., glucose transport, GLUT4 mRNA). This was subsequently extended into whole animal studies and clinical trials.

Throughout we use the scientific and common names of the medicinal plants, and describe the chemistry used in the extraction process–a key consideration for studies of this type. In all tables, extracts are alphabetized by species unless multiple different species were used in the same study, in which case these are placed arbitrarily at the top of each table for clarity.

Results

Studies using cell lines

Effects of medicinal plants and phytochemicals on glucose transport and glucose transporters

Defective insulin-stimulated glucose transport is hallmark of T2DM (Ogurtsova et al., 2017; Nandabalan, Sujatha & Shanmuganathan, 2010; Drissi et al., 2021; Stadlbauer et al., 2016). Glucose transporters (GLUT) of the facilitative diffusion type are a multi-gene family of proteins which function to move glucose across cell membranes (Ogurtsova et al., 2017; World Health Organization, 2019; Singhal, Bangar & Naithani, 2012). Among the facilitative GLUT isoforms, GLUT4 is particularly important as it is expressed predominantly in skeletal and adipose tissues and accounts for post-prandial glucose disposal in these tissues (World Health Organization, 2019; Nandabalan, Sujatha & Shanmuganathan, 2010). Skeletal muscle contribute largely to a greater part of the total body mass in humans and it regulates several physiological processes including up to 85% of insulin-mediated glucose up-take through GLUT4 (Ahmad, Choi & Lee, 2020). Many studies have utilized this for therapeutic management of diabetes and in particular the role of extracellular matrix (Ahmad, Choi & Lee, 2020). Skeletal muscle contraction during exercise improves GLUT4 translocation to the cell membrane for glucose uptake and insulin-sensitivity (Jiang et al., 2013). Also, altered muscle glycogen synthesis play a major role in insulin resistance, and glycogen synthase, hexokinase, and GLUT4 are the major culprit involved in the skeletal muscle pathogenesis of type 2 dibetes (Petersen & Shulman, 2002; Saini, 2010). GLUT2 and GLUT5 are responsible for intestinal glucose and fructose uptake (Schreck & Melzig, 2021), while GLUT1 is present ubiquitously in all the body tissues (Galochkina et al., 2019). The dominant glucose transporters found in the small intestine are sodium-glucose linked transporter 1 (SGLT1) which accumulates glucose into adsorptive epithelial cells against its concentration gradient and GLUT2 which mediates movement of glucose from the epithelial cells into the blood (Schreck & Melzig, 2021); inhibition reduces the amount of glucose absorbed into the body. The hemodynamic activities of the glucose transporters have been extensively researched (Ogurtsova et al., 2017; World Health Organization, 2019; Singhal, Bangar & Naithani, 2012; Nandabalan, Sujatha & Shanmuganathan, 2010). Medicinal plants’ products and phytochemicals that modify the action of the glucose transporters could significantly contribute to the search for effective drugs in the management of diabetes. Table 1 is a collation of studies of medicinal plants known to modulate glucose transport in cell lines. Some notable highlights of this extensive literature are discussed briefly below.

Table 1 Medicinal plant active on glucose transporters.

Shown are studies in which the indicated plants have been shown to drive a change in glucose transport. Where possible these have been alphabetized, but studies in which multiple plant species or extracts were used in a single study are shown at the top of the table.

Medicinal plant	Phytochemistry	Key effectors	Summary	References	
Aronia melanocarpa, Cornus officinalis, Crataegus pinnatifida, Lycium chinense, Vaccinium myrtillus, Brassica oleracea, Juglans regia, Peumus boldus, Adenophora triphylla, Eucommia ulmoides, and Malus domestica	Methanolic extract of the leaves, roots, aqueous extract from the bark, and fruit skin.	SGLT 1 and GLUT2.	Inhibition of intestinal SGLT1 and GLUT2 in Caco-2 cells.	(Schreck & Melzig, 2021)	
Hoodia, Sapindus mukorossi, Quillaja saponaria, Papaver, Castanea, Bitter orange, Oregon grape, Saposhnikovia divaricata, Sponge gourd, Black radish, Asparagus, Neem, Uzara, Reetha B, Chelidonium majus, Teasel, Tetradium ruticarpum, Southern wax myrtle, Bistort, Indian tobacco, Figwort, Rangoon creeper, Peruvian rhatany, Chinese rhubarb, Poppy capsule and flowers, Ivy, Common daisy leaves and flowers, Rosebay willowherb, and Goldenrod.	Plant extracts.	GLUT4	Stimulation of GLUT4 translocation in CHO-K1 and 3T3-L1 cells and plasma membrane insertion of GLUT4 in Hela cells.	(Stadlbauer et al., 2021)	
Trigonella foenumgraecum, Urtica dioica, Atriplex halimus, and Cinnamomum verum	50% ethanol extract of the various parts.	GLUT4	Increased translocation of GLUT4 to the plasma membrane in L6-GLUT4myc rat muscle cells.	(Kadan et al., 2013)	
Rhododendron groenlandicum, Alnus incana, Sarracenia purpurea	Leaf, bark, and whole plant, respectively.	GLUT4	Increased total membrane expression of GLUT4 and phosphorylation of AKT and AMP in C2C12 and H4IIE cell lines.	(Shang et al., 2015)	
Strawberry and Apple	Polyphenols, phenolic acid, and tannins.	GLUT2, SGLT1	Inhibition of GLUT2 and SGLT1 in human intestinal Caco-2 cells.	(Manzano & Williamson, 2010)	
Annona stenophylla	Aqueous root extract.	GLUT4	Enhanced GLUT4 and gene expression in C2C12 muscle cell lines.	(Taderera et al., 2019)	
Apios americana	Glycosides from the leaves.	MAPK and glucose uptake	Restores glucose uptake, glucose consumption, and glycogen content in HepG2 cells via MAPK and Nrf2 pathways.	(Yan et al., 2017)	
Capparis moonii	Gallotannins from hydro-alcoholic fruit extract.	GLUT4	Increased phosphorylation of IR-β, IRS-1, and GLUT4, PI3K mRNA expression in L6 myotube cells.	(Kanaujia et al., 2010)	
Cassia abbreviate	Aqueous leaf, seed, and bark extract.	GLUT4	Enhanced GLUT4 translocation and gene expression in C2C12 mouse skeletal muscle cells.	(Kamatou, Ssemakalu & Shai, 2021)	
Cinnamomum burmannii	Water extract and polyphenols.	GLUT4
GLUT1	Increased expression of mRNA GLUT4, IR, GLUT1in mouse 3T3- adipocytes.	(Cao, Polansky & Anderson, 2007; Cao, Graves & Anderson, 2010)	
Cinnamomum cassia	Cinnamic acid from a hydroalcoholic bark extract.	GLUT4	Increased GLUT4 mRNA and inhibition of PTP1B activity in L6 myotubes.	(Lakshmi et al., 2009)	
Citrullus colocynthis	Fruit and seed extracts and solvent fractions.	GLUT4	Enhancement of insulin-induced GLUT4 translocation in adipocytes.	(Drissi et al., 2021)	
Costus igneus (insulin plant)	Leaf extract	Glucokinase/GLUT2	Increased glucokinase activity, insulin, and GLUT2 gene expression but inhibition of glucose-6-phosphatase activity in human hematopoietic stem cells (HSCs) showing β-like cells action.
C. igneus contained insulin-like proteins (ILP) with hypoglycemic activities in insulin-responsive cell line RIN 5f.	(Kattaru et al., 2021; Joshi et al., 2013)	
Dandelion powder	Chloroform extract.	GLUT4	Increased GLUT4 expression and membrane translocation via the AMPK pathway in L6 cells.	(Zhao et al., 2018b)	
Folium sennae	Ethanol extract.	GLUT4	Promotes membrane translocation and mRNA of GLUT4 via AMPK, AKT, and G protein-PLC-PKT pathways and internalization of C2+ in L6 cells.	(Zhao et al., 2018a)	
Gundelia tournefortii	Hexane and methanol extract of the aerial part.	GLUT4	Enhanced translocation of GLUT4 to the plasma membrane by the methanol extract than the hexane extract in L6 myotube cells.	(Kadan et al., 2018)	
Kigelia pinnata	Isolated phytochemicals from ethanol extract of K. pinnata twigs.	GLUT4	Increased GLUT4 translocation to the skeletal muscle cell surface in skeletal muscle cells.	(Faheem et al., 2012)	
Mangifera indica	Ethylacetate extract and 3β-taraxerol.	GLUT4	GLUT4 translocation and glycogen synthesis in 3T3-L1 adipocytes.	(Nandabalan, Sujatha & Shanmuganathan, 2010)	
Maydis stigma [corn silk]	Extracted polysaccharides.	GLUT4	Membrane translocation of GLUT4 in rats L6 skeletal muscle and regulation of PI3K/AKT pathways.	(Guo et al., 2019)	
Mitragyna speciosa	Water, methanol extract, and mitragynine [a principal constituent].	GLUT1	Increased GLUT1 content in rat L6 myotubes.	(Purintrapiban et al., 2011)	
Momordica balsamina	ethanol, ethyl acetate, and n-hexane fruit extract	GLUT2	Increased GLUT2 gene expression	(Kgopa, Shai & Mogale, 2020)	
Momordica charantia	Aqueous and chloroform extract of the fruit.	GLUT4	Increased glucose uptake with GLUT4, PPARγ, and PI3K mRNA gene expression in L6 myotube cells.	(Kumar et al., 2009)	
Moringa concanensis	Leaf extract	GLUT4 via PPARγ effects	3T3-L1 adipocytes, enhanced GLUT4 gene expression	(Balakrishnan, Krishnasamy & Choi, 2018)	
Morus alba	Ethanol leaf extract.	GLUT4	Stimulation of glucose uptake and GLUT4 translocation to the plasma membrane via activation of PI3K in rat adipocytes.	(Naowaboot et al., 2012)	
Nymphaea nouchali	Seed extracts	GLUT4 via PPARγ effects	Increased GLUT4 mRNA expression	(Parimala et al., 2015)	
Ocimum basilicum	Methanol, hexane, and dichloromethane are extracts of the stem, leaf, and flowers.	GLUT4	Elevated GLUT4 translocation to the plasma membrane HepG2 and rat L6 muscle cells.	(Kadan et al., 2016)	
Panax ginseng [black ginseng]	Ethanolic extract of black ginseng.	GLUT4	Increased phosphorylation of AMPK, increased upregulation of GLUT2 in the liver and GLUT4 in the muscle.	(Kang et al., 2017)	
Pinus pinea [pine]	Bark extract	MAPK and glucose uptake	Activation of p38MAPK, which in turn activates SGLT1 and GLUT2 in Caco-2 cells.	(El-Zein & Kreydiyyeh, 2011)	
Portulaca oleracea and Coccinia grandis	Plant extract.	GLUT4	PI3K mediated GLUT4 translocation in insulin-sensitive CHO-K 1 cells and adipocytes.	(Stadlbauer et al., 2016)	
Rosemary	Carnosol [diterpene] found in Rosemary.	GLUT4	AMP-dependent increase GLUT4 translocation in L6 skeletal muscle cells.	(Vlavcheski et al., 2018)	
Salacia oblonga	Hot water extract of the root, stem, and mangiferin, the bioactive compound.	GLUT4	GLUT4 and concomitant phosphorylation of 5’AMP-activated protein kinase in L6 myotubes and 3T3- adipocytes.	(Giro et al., 2009)	
Selaginella tamariscina	Selaginellins and bioflavonoids from methanol extract.	PTP1B and glucose uptake	Glucose uptake and inhibition of PTP1B in 3T3-L1 adipocytes	(El-Zein & Kreydiyyeh, 2011; Giro et al., 2009; Nguyen et al., 2015a)	
Sinocrassula indica Berge	Ethanolic extract	GLUT1, GLUT4	Increased glucose uptake in L6 myotubes and H4IIE hepatoma cells	(Yin et al., 2009)	
Gymnema sylvestre	Methanolic leaf extract	GLUT4	Enhanced glucose uptake in L6 myotubes cells	(Kumar et al., 2016)	

In one of the most comprehensive studies, Schreck & Melzig (2021) used Caco-2 cells exposed to a range of plant extracts to identify potential inhibitors of glucose transport. They reported between 40% to 80% reduction using the methanolic extracts of a range of plants including the fruits of Aronia melanocarpa, Valcheva-Kusmanova et al. (2007) Cornus officinalis, Crataegus pinnatifida, Lycium chinense, and Vaccinium myrtillus; the leaves of Brassica oleracea, Juglans regia, Peumus boldus, and the roots of Adenophora triphylla. The authors also reported 50% to 70% reduction by aqueous extract from the bark of Eucommia ulmoides and fruit skin of Malus domestica. These effects are likely acting via inhibition of GLUT1, the predominant transporter in these cells.

One of the key facets of insulin action is to drive the delivery of GLUT4 molecules from intracellular stores to the surface of fat and muscle cells, a process called ‘translocation’. Stadlbauer et al. (2021) used CHO-K1 cells expressing GLUT4 and total internal reflection microscopy to identify GLUT4 translocation-inducing effects of some thirty plant extracts. Though the taxonomy of some of the plants were not fully defined, they included Hoodia, Sapindus mukorossi, Quillaja saponaria, Papaver, Castanea, Bitter orange (genus and species not specified), Oregon grape (genus and species not specified), Common daisy flowers, Rosebay willowherb leaves and Goldenrod flower as potential compounds that could be exploited as potential anti-hyperglycemic agents in the treatment of T2DM via effects on GLUT4 redistribution.

While the above study used a non-classical insulin target tissue (for good experimental reasons), others have focused upon more physiological cell systems. Through bioassay-guided fractionation, Kanaujia et al. (2010) reported two chebulinic acid derivatives from Capparis moonii with significant stimulatory effects on glucose uptake effects concomitant with increased IR-β, Insulin receptor substrate-1 (IRS-1) phosphorylation, and mRNA expression of GLUT4 and PI3K in L6 muscle cells. Carnosol from rosemary extract stimulated AMPK-dependent GLUT4 translocation with no effect on Akt phosphorylation in L6 myotubes (Vlavcheski et al., 2018). Methanolic extract of Gundelia tournefortii potentiated insulin-stimulated GLUT4 translocation to the plasma membrane in skeletal muscle L6 cells (Kadan et al., 2018). An aqueous extract of Cassia abbreviata induced a two-fold increase in GLUT4 translocation in C2C12 (mouse) skeletal muscle cells probably via activation of the canonical PI3K/Akt pathway (Kamatou, Ssemakalu & Shai, 2021).

Naowaboot and colleagues reported the mechanism of antihyperglycemic effects of Morus alba leaf extract, including increasing glucose uptake via activation of the PI3K pathway and the plasma membrane translocation of GLUT4 in rat adipocytes (Naowaboot et al., 2012). Ethyl acetate extract and 3β-taraxerol isolated from Mangifera indica promoted increased GLUT4 translocation and glycogen synthesis in 3T3-L1 adipocytes (Nandabalan, Sujatha & Shanmuganathan, 2010). The study also noted the effect on glycogen synthesis was due to PI3K-dependent activation of Akt with subsequent inactivation of glycogen synthase kinase 3B (GSK3β) phosphorylation (discussed further below). The fruit of Citrullus colocynthis enhanced insulin-induced GLUT4 translocation and Akt phosphorylation in 3T3-L1 adipocytes (Drissi et al., 2021).

Stadlbauer et al. (2016) screened further natural products as alternatives to insulin through quantitation of GLUT4 translocation in insulin-sensitive CHO-K1 and importantly extended their remarkable study into commercial adipocyte cells. Of the seven medicinal plants tested, Portulaca oleracea and Coccinia grandis were found to induce GLUT4 translocation together with increased glucose concentration uptake likely mediated by PI3K/Akt pathway in adipocytes (Stadlbauer et al., 2016).

Such findings, together with the many others noted in Table 1 and which space preclude detailed discussion of here, suggest natural products can drive re-distribution of GLUT4 to the plasma membrane in an insulin-mimetic manner. However, the effects are not confined to GLUT4. For example, Kang and colleagues reported upregulation of GLUT2 in the liver (and up-regulated GLUT4 in muscle) as the possible mechanism of the antidiabetic effects of Panax ginseng (black ginseng). Manzano & Williamson (2010) investigated the glucose uptake inhibition of polyphenols, phenolic acid, and tannins from strawberry (var. Abion) and apple (var. golden delicious) in Caco-2 intestinal cell monolayers and reported increased inhibition of GLUT2 and SGLT1 and reduced glucose intestinal bilayer transport. Enhancement of glucose uptake by mitragyna speciosa and mitragynine in rat L6 myotubes is associated with increased GLUT1 protein content (Purintrapiban et al., 2011). And the glucose uptake (GLUT4) enhancement activity of G. sylvestre in L6 myotubes alongside the amelioration of insulin resistance in the 3T3-L1 adipocytes cells have been reported (Kumar et al., 2016). One further study of note is the report that the methanol extract of the aerial part of Selaginella tamariscina enhanced glucose uptake in 3T3-L1 adipocytes, possibly by inhibition of PTP1B (El-Zein & Kreydiyyeh, 2011; Giro et al., 2009). Collectively, these studies reveal that medicinal plants have a range of action on glucose transport across multiple tissues of relevance to the treatment of diabetes.

Medicinal plants and phytochemical effects on PI3K and Akt activity

In the canonical insulin signaling pathway controlling glucose homeostasis, insulin receptor substrates are phosphorylated on tyrosine residues which then act as docking sites for downstream signaling molecules. Of particular note, IRS-1 recruits phosphoinositide 3-kinase (PI3K) (Petersen & Shulman, 2018). PI3K phosphorylates phosphatidylinositol 4,5-biphosphate to form phosphatidyl-inositol 3,4,5-trisphosphate that in turn promotes Akt phosphorylation and activation (World Health Organization, 2019; Luna-Vital & De Mejia, 2018). Akt is an important nexus on the insulin signaling cascade because of its multi-substrate activities (Naowaboot et al., 2012; Nandabalan, Sujatha & Shanmuganathan, 2010; Tonks et al., 2013; Huang et al., 2018). Phosphorylation of Akt initiates a cascade of downstream events through many substrates, including phosphorylation of Akt substrate of 160 kDa (AS160), a RabGAP (GTPase activating protein); this in turn leads to GLUT4 translocation in muscle and adipose cells (Matschinsky, 2005; Sharma et al., 2021). Defects in Akt phosphorylation, as seen in impaired insulin activation, are associated with development of muscle and adipose insulin resistance in obesity and T2DM (Naowaboot et al., 2012; Joshi et al., 2013; Luna-Vital & De Mejia, 2018; Kim et al., 1999). The insulin PI3K/Akt pathway has been widely targeted in T2DM pharmacotherapy (World Health Organization, 2019; Naowaboot et al., 2012; Luna-Vital & De Mejia, 2018) and is important in the pathophysiology and therapy of other diseases (Nurcahyanti et al., 2021). Evidence of medicinal plants and phytochemicals modifying PI3K/Akt actions have been documented and are presented in Table 2. Some examples are briefly outlined below.

Table 2 Medicinal plants modifying PI3K/Akt activity.

Shown are studies in which the indicated plants have been shown to drive activation of the canonical insulin signalling molecules PI3K and Akt. Table is constructed in alphabetical order of plant species.

Medicinal plant	Phytochemistry	Target	Summary	References	
Anemarrhena asphodeloides Bunge	Monosaccharides	PI3K	Activation of PI3K/Akt, IRS-1 signaling pathway, and inhibition of α-glucosidase activity in HepG2 cells.	(Chen et al., 2022)	
Broussonetia kazinoki	A flavan [Kazinol B] purified the root.	Akt	Improved insulin sensitivity via Akt and AMPK activation in 3T3-L1 adipocytes.	(Lee et al., 2016)	
Dendrobium officinale	Polysaccharide	PI3K	Increased PI3K/Akt phosphorylation in insulin-resistant HepG2 cells.	(Wang et al., 2018)	
Folium sennae	Ethanol extract.	Akt	Increased AMPK, Akt, and PKC phosphorylation in L6 rat skeletal muscle.	(Zhao et al., 2018a)	
Grifola frondosa	Heteropolysaccharide of the fruiting body.	PI3K	Increased mRNA of IRS1/PI3K, and downregulation of JNK1 signaling in HepG2 cells.	(Chen et al., 2018)	
Juniperus chinensis	α- Methyl artoflavano coumarin from Juniperus chinensis.	Akt	PI3K/Akt and inhibition of PTP1B in HepG2 cells.	(Jung et al., 2017)	
Mangifera indica	Ethyl acetate extract [EAE] and 3β-taraxerol phytochemistry.	PI3K	Increased PI3K level and GLUT4 translocation in 3T3-L1 adipocytes.	(Nandabalan, Sujatha & Shanmuganathan, 2010)	
Maydis stigma [corn silk]	Maydis stigma [corn silk] extract.	Akt	A dose-dependent increase in expression of p-Akt/Akt in L6 skeletal muscle myotubes.	(Guo et al., 2019)	
Nigella glandulifera	Alkaloids from the seeds.	PI3K	Increased PI3K/Akt pathway together with inhibition of PTP1B.	(Tang et al., 2017)	
Nigella glandulifera	Norditerpenoid alkaloids of the seeds.	Akt	PI3K/Akt pathway, inhibition of PTP1B, increased glycogen synthesis with hexokinase activity in L6 myotubes	(Tang et al., 2017)	
Sargassum pallidum	Homogeneous polysaccharides.	PI3K	Upregulation of PI3K, GS, and IRS-1 gene expression in insulin-resistant HepG2 cells.	(Cao et al., 2019)	
Zhenjiang aromatic vinegar	Polyphenol-rich extract.	PI3K	Activation of PI3K/Akt pathway in IR-HepG2 cells.	(Xia et al., 2021)	

An ethyl acetate extract and 3β-taraxerol of Mangifera indica significantly activate GLUT4 translocation via a PI3K dependent pathway in 3T3-L1 adipocytes (Nandabalan, Sujatha & Shanmuganathan, 2010). Polysaccharides from corn silk (Maydis stigma) increased phosphorylation of Akt in a dose-dependent manner in L6 skeletal muscle myotubes (Guo et al., 2019). Similarly, phosphorylation of Akt in response to an extract of Folium sennae was described, together with a significant enhancement of GLUT4 translocation (Zhao et al., 2018a). Antidiabetic effect of alkaloids from the seeds of Nigella glandulifera increased the activity of the PI3K/Akt pathway in L6 myotubes with a concomitant increase in glycogen synthesis and hexokinase activity (Tang et al., 2017). Four C21 steroidal glycosides A-D from G.sylvestre promoted GLUT4 translocation to the plasma membrane in L6 cells via activation of PI3K/AKT (Li et al., 2019a). These and other studies (Table 2) point to a significant number of useful PI3K/Akt modulators within medicinal plants.

Many studies have examined effects of medicinal plants in hepatoma cells, as liver is a key site of post-prandial glucose disposal with a notable emphasis on PI3K/Akt signaling (Table 2). Heteropolysaccharides of Grifola frondosa (edible mushroom) increased IRS1/PI3K mRNA levels and enhanced insulin sensitivity (Chen et al., 2018). Polysaccharides of Dendrobium officinale increased the activity of PI3K and Akt and partially ameliorated symptoms in diabetic mice, pointing to a clear effect in a complex organism rather than simply in cell culture models (Wang et al., 2018). Inhibition of phosphorylated insulin receptor substrate-1(IRS-1), but activation of PI3K/Akt in insulin-resistant HepG2 cells by polyphenol-rich extract of Zhenjiang aromatic vinegar has been documented (Xia et al., 2021). Homogeneous polysaccharides from Sargassum pallidum ameliorate insulin resistance by upregulation of PI3K, Glycogen synthase (GS), and IRS-1 expression in insulin-resistant HepG2 cells (Cao et al., 2019). Monosaccharides from Anemarrhena asphodeloides Bunge exhibited hypoglycemic effects by activating PI3K/Akt, IRS-1signaling pathway, and inhibiting α-glucosidase activities in insulin-resistant-HepG2 cells (Chen et al., 2022). A natural flavonocoumarin (α-Methylartoflavonocoumarin) isolated from Juniperus chinensis was reported to activate PI3K/Akt pathway in insulin-resistant HepG2 cells (Jung et al., 2017). Thus, effects are evident both at the level of gene expression and activity mediated by a range of extracts.

Activity of medicinal plants and phytochemicals on glucokinase

Glucokinase (GCK) is important in the regulation of glucose metabolism in liver and pancreas. GCK is essential for pancreatic insulin secretion and hepatic insulin action via phosphorylation of glucose to glucose 6-phosphate (Naowaboot et al., 2012; Balakrishnan, Krishnasamy & Choi, 2018). Low GCK levels have been observed in T2DM (Haeusler et al., 2015) and could serve as a potential drug target for therapeutic intervention (Kim et al., 2013; Yang, Jang & Hwang, 2012). GCK is currently being targeted as therapeutic for T2DM (Matschinsky, 2009; Matschinsky & Wilson, 2019). Glucokinase activators have been advocated as an alternative approach to restoring and improving glycemic control in T2DM (Zhou et al., 2001; Towler & Hardie, 2007; Toulis et al., 2020). Several medicinal plants and phytochemicals with glucokinase activities have been recognized (Sharma et al., 2021). We have focused on the cellular and molecular glucokinase activities of medicinal plants in tissue culture models (Table 3).

Table 3 Medicinal plants with activities on glucokinase (GCK).

Shown are studies in which the indicated plants have been shown to drive changes in glucokinase activity or expression. Table is constructed in alphabetical order of plant species.

Medicinal plant	Phytochemistry	Summary	References	
Costus igneus (insulin plant)	Leaf extract.	Increased glucokinase activity, insulin, and GLUT2 gene expression but inhibition of glucose-6-phosphatase activity in human hematopoietic stem cells (HSCs) showing β-like cells action.
C. igneus contained insulin-like proteins (ILP) with hypoglycemic activities in insulin-responsive cell line RIN 5f.	(Kattaru et al., 2021; Joshi et al., 2013)	
Momordica balsamina	Ethanol, ethyl acetate, and n-hexane fruit extract.	Increased GCK and GLUT2 mRNA gene expression in RIN-m5F cells.	(Kgopa, Shai & Mogale, 2020)	
Zea mays (Purple corn)	Anthocyanins from the pericarp.	Activation of GCK in HepG2 cells decreased glucose uptake in Caco-2 cells and increased glucose-stimulated insulin secretion in iNS-1E in the pancreas.	(Luna-Vital & De Mejia, 2018)	

Glucokinase activation by the leaf extract of Costus igneus (known in India as the ‘insulin plant’ for its purported anti-diabetic action) was examined in differentiated human hematopoietic stem cell (HSCs) as a model of β-cells. The extract increased GCK and inhibited of glucose-6-phosphatase activity by C. igneus, thereby improving glucose sensing, insulin production, and decreased gluconeogenesis (Kattaru et al., 2021). It was also reported that C. igneus profoundly increased insulin receptor and GLUT2 gene expression (Table 1). Insulin-like proteins (ILP) purified from C. igneus also showed hypoglycemic activity in insulin-responsive cell line RIN 5f cells (Joshi et al., 2013). Activation of free fatty acid-receptor1 (FFAR1) and GCK by anthocyanin-rich extract from the pericarp of purple corn was demonstrated in HepG2 cells (Luna-Vital & De Mejia, 2018). Significant elevations in glucokinase gene expression in response to ethanol, ethyl acetate, and n-hexane fruit extract of Momordica balsamina, were reported in RIN-m5F cells (Kgopa, Shai & Mogale, 2020).

Medicinal plants modifying activity of glycogen synthase kinase-3 (GSK-3)

GSK-3 inhibits glycogen synthase activity. Insulin phosphorylates GSK-3 and prevents glycogen synthase inactivation (Nabben & Neumann, 2016). This role of GSK-3 in the insulin signaling pathway provides a mechanistic approach to the use of GSK-3 inhibitors in the treatment of insulin-resistant diabetes. Two studies are worthy of comment. Ethanolic extract of Shilianhua (Sinocrassula indica Berge) was found to induce GSK-3β phosphorylation similarly to insulin in 3T3-L1 preadipocytes and rat skeletal L6 myoblasts, indicating a possible mechanism of antidiabetic activity (Yin et al., 2009). This extract also enhanced insulin-stimulated glucose consumption in L6 myotubes and H4IIE hepatocytes, and insulin-independent glucose uptake in 3T3-L1 adipocytes. The result also showed increased GLUT1 protein expression in 3T3-L1 and GLUT4 protein expression in L6 myotubes cells (Yin et al., 2009). Hot water reduction from the root of Sarcopoterium spinosum increased glycogen synthesis via induction of GSK-3 β phosphorylation in L6 myotubes (Sahuc, 2016). S. spinosum also enhanced basal insulin secretion in the pancreatic β-cells and inhibited isoproterenol-induced lipolysis in 3T3-L1 adipocytes (Sahuc, 2016).

Peroxisome proliferator-activated receptor-gamma (PPARγ) and medicinal plants

Peroxisome proliferator-activated receptor gamma (PPARγ) is a member of the nuclear receptor super-family which play integral roles in glucose and lipid metabolism (Mirza, Althagafi & Shamshad, 2019). These receptors are targets for diabetes therapy and also for the treatment of cardiovascular disease, cancer, and inflammation (Mirza, Althagafi & Shamshad, 2019). We present the effects of various medicinal plants on PPARγ activity and gene expression (Table 4).

Table 4 Medicinal plants modifying activities of peroxisome proliferator-activated receptor-gamma (PPARγ).

Shown are studies in which the indicated plants have been shown to mediate effects probably via PPRγ. Table is constructed in alphabetical order of plant species. Where possible these have been alphabetized, but studies in which multiple plant species or extracts were used in a single study are shown at the top of the table.

Medicinal plant	Phytochemistry	Summary	References	
Yeongyang korea (Korea red pepper),
Capsicum annuum	Ethanol extract.	Increased PPARγ and AMPK phosphorylation in C2C12 myotubes.	(Yang, Jang & Hwang, 2012)	
Boehmeria nivea	Ethanol leaf extract.	Increased mRNA levels of PPARγ in C2C12 myotubes cells.	(Kim et al., 2013)	
Miconia sp.	Ethanol extract of the aerial part.	Increased PPARγ mRNA and GLUT4 in 3T3-L1 adipocytes.	(Ortíz-Martinez et al., 2016)	
Momordica charantia	Chloroform extract of the fruit.	Increased mRNA gene expression of PPARγ in L6 myotube skeletal muscle cells, as well as GLUT4 and PI3K.	(Kumar et al., 2009)	
Moringa concanensis	Leaf extract.	Upregulation of mRNA of PPARγ, GLUT4, FAS, Tsrebp, DAG, and Akt signaling in 3T3-L1 adipocytes.	(Balakrishnan, Krishnasamy & Choi, 2018)	
Nymphaea nouchali	Seed extract.	Increased mRNA of PPARγ and GLUT4 in 3T3-L1 adipocytes.	(Parimala et al., 2015)	
Punica granatum	Flower aqueous extract and ethyl acetate fraction.	Increased mRNA PPARγ gene and protein expression in TPH-1-derived macrophage cell line.	(Huang et al., 2005)	

Chloroform extract of the fruit of Momordica charantia has been reported to significantly increase PPARγ gene expression 2.8-fold, comparable to the insulin sensitizer rosiglitazone (2.4-fold) in L6 myotube skeletal muscle cells (Kumar et al., 2009). Huang et al. (2005) demonstrated that Punica granatum flower extract and ethyl acetate fractions enhanced PPARγ gene expression and protein levels in a macrophage cell line. Increased mRNA of PPARγ (and GLUT4) by Nymphaea nouchali seed extract in 3T3-L1 adipocytes as the possible mechanism of its anti-hyperglycemic effect was reported (Parimala et al., 2015). Exposure of 3T3-L1 adipocytes to an ethanol extract of Miconia increased mRNA of PPARγ by 1.4-fold and inhibited α-amylase and α-glucosidase. The extract also increased lipid accumulation by around 30% as a possible anti-diabetic mechanism of action (Ortíz-Martinez et al., 2016). Upregulation of PPARγ together with GLUT4, SREBP and FAS expression was observed in 3T3-L1 adipocytes treated with the leaf extract of Moringa concanensis (Balakrishnan, Krishnasamy & Choi, 2018). Effects in muscle models have also been reported: Kim et al. (2013) reported increased transcription activity and mRNA levels of PPARγ in C2C12 myotubes by Boehmeria nivea ethanol leaf extract and Korean red peppers (Yeongyang korea) increased glucose uptake in C2C12 via increased transcriptional activity of PPARγ (Yang, Jang & Hwang, 2012).

AMP-activated protein kinase (AMPK)

AMP-activated protein kinase (AMPK) is a known energy sensor for metabolic homeostasis (Steinberg & Carling, 2019) which plays a central role in regulating lipid and protein metabolism together with fatty acid oxidation and muscle glucose uptake (Sathishsekar & Subramanian, 2005). AMPK plays a crucial role in insulin sensitivity, which explains its place as a potential drug candidate for T2DM therapy (Tasic et al., 2021; Hawkins et al., 2021). AMPK systems have been said to be partly responsible for the health benefits of exercise and AMPK is an important downstream effector of metformin. It has also been proposed as a possible target for novel drugs in managing obesity, type 2 diabetes, and metabolic syndrome (Hu, Zeng & Tomlinson, 2014; Hosseini et al., 2014; Kim et al., 2016). Ethnopharmacological investigators have reported that several medicinal plants modulate the activity of AMPK in cell models, and we have summarized these reports in Table 5, and highlight a few notable studies below.

Table 5 Medicinal plants regulating AMP-activated protein kinase (AMPK).

Shown are studies in which the indicated plants have been shown to mediate effects via AMPK activity. Table is constructed in alphabetical order of plant species.

Medicinal plant	Phytochemistry	Summary	References	
Artemisia dracunculus	Alcoholic extract (PMI-5011).	Increased insulin secretion through AMPK activation in NIT-1 cells.		
Artemisia sacrorum	Petroleum ether fraction.	Decreased glucose production via the AMPK-GSK-CREB pathway in HepG2 cells.	(Yuan & Piao, 2011)	
Aspalathus linearis	80% ethanol extract.	Amelioration of insulin resistance in C2C12 via activation of AMPK and Akt pathway.	(Mazibuko et al., 2013)	
Cimicifuga racemosa	Ethanol extract and Phyto-compounds.	Increased AMPK activity in HepaRG cells.	(Moser et al., 2014)	
Crocus sativus [Saffron]	Methanol extract.	Increased glucose uptake and insulin sensitivity via AMPK phosphorylation in C2C12 mouse myotubes cells.	(Kang et al., 2012)	
Entada phaseoloides	Total saponin extract.	Suppression of hepatic gluconeogenesis via AMPK and Akt/GSK3β in Primary hepatocytes and HepG2 cells.	(Zheng et al., 2016)	
Iris sanguinea	Isolated compounds from methanol extract of the seeds.	Increased glucose uptake via activation of ACC and AMPK in mouse C2C12 skeletal myoblast.	(Yang et al., 2017)	
Malva verticillata	Ethanol extract and compound isolate [β-sitosterol].	Increased glucose uptake via AMPK phosphorylation in L6 myotubes.	(Jeong & Song, 2011)	
Momordica charantia	Triterpenoids from the stem.	Overcome insulin resistance via AMPK activation in FL83B and C2C12 cells.	(Cheng et al., 2008)	
Psidium guajava	Flavonoids from the leaves.	AMPK phosphorylation in rat L6 myotubes and L02 human hepatic cells.	(Li et al., 2019b)	
Rhodiola crenulata	Methanol extract.	Inhibition of gluconeogenesis in human hepatic HepG2 cell via activation of AMPK.	(Lee et al., 2015)	
Rosmarinus officinalis	Dichloromethane-methanol extract.	Regulate glucose and lipid metabolism through activation of AMPK and PPAR pathways in HepG2 cells.	(Tu et al., 2013)	
Sechium edule	Water and polyphenol extract of the shoot.	Inhibition of lipogenesis and stimulation of lipolysis via AMPK activation and decreased lipogenic enzymes in HepG2 cells.	(Wu et al., 2014, 2020)	
Stauntonia chinensis	Triterpenoid saponins.	Increased glucose uptake in HepG2 insulin-resistant cells via AMPK phosphorylation and IR, IRS-1, PI3K/Akt pathways	(Hu et al., 2014)	
Toona sinensis	Leaf extract.	Increased glucose uptake in C2C12 myotubes due to AMPK activation.	(Liu et al., 2015)	
Vigna angularis (Azuki bean)	Extract	Increased phosphorylation of AMPK and Akt in HepG2 cells.	(Sato et al., 2016)	

Ethanolic extract and phytochemical compounds from Cimicifuga racemosa mediate increased AMPK activity in fully differentiated HepaRG cells and is a possible mechanism of antidiabetic activity (Moser et al., 2014). Yuan & Piao (2011) reported activation of AMPK by the petroleum ether fraction of Artemisia sacrorum in HepG2 cells. They showed increased phosphorylation of AMPK (on T172), acetyl-CoA carboxylase (ACC; reside S79), and GSK-3β and reported concomitant downregulation of phosphoenolpyruvate carboxykinase (PEPCK), and glucose-6-phosphatase (G6Pase). Similarly, Zheng et al. (2016) exploring the anti T2DM activity of Entada phaseoloides in primary mouse hepatocytes and HepG2 cells, reported suppression of hepatic gluconeogenesis via activation of the AMPK signaling pathway and Akt/GSK-3β. Extract of Rosmarinus officinalis significantly increased glucose consumption in HepG2 cells via increased phosphorylation of AMPK and ACC and potentially increased liver glycolysis and fatty acid oxidation (Tu et al., 2013). Thus, numerous examples of medicinal plants exerting effects via AMPK in hepatoma cell lines have been described (Table 5).

Effects mediated via AMPK have been reported in other cell types. These include an alcoholic extract of Artemisia dracunculus enhanced insulin release from β-cells isolated from mouse and human islets via activation of AMPK and suppressed LPS/IFNγ-induced inflammation. Effects on glucose transport in muscle lines include an extract of Crocus sativus (saffron) which increased glucose uptake and insulin sensitivity in C2C12 myotubes by increased phosphorylation of AMPK in a dose and time-dependent manner (Kang et al., 2012) and compounds isolated from the seed of Iris sanguinea was reported to be via AMPK and ACC phosphorylation in the same cell type (Yang et al., 2017).

Ethanolic extract and isolated compound (β-sitosterol) from Malva verticillata seed significantly increased activation of AMPK as the molecular mechanism for glucose uptake in L6 myotubes (Jeong & Song, 2011). Triterpenoids from the stem of Momordica charantia have been reported to overcome insulin resistance in FL83B and C2C12 via AMPK activation (Cheng et al., 2008). Hence, the effects of such compounds on AMPK is an active and vigorous area of research.

Studies in animal models

The process of drug development encompasses pre-clinical experimentation (in vitro, in silico, and in vivo) leading ultimately to clinical trials in humans. To understand if there is ongoing vertical research towards developing antidiabetic agents from these medicinal plants, we reviewed their exploitation in experimental animal models as summarized in Table 6. Some highlights are discussed below.

Table 6 Medicinal plants having antidiabetic activity in tissue culture and whole animal biology.

Studies of medicinal plants with demonstrated anti-diabetic properties are listed. Plants are arranged in alphabetical order. The animal model studies are cross-referenced to cellular studies of the same extract/plant wherever possible.

Medicinal plants	Phytochemistry	Animal model	Summary	Animal
study	Cell study	
Yeongyang korea (Korea red pepper),
Capsicum annuum	Seed extract.	Mice	Improved glycemic control, decreased hepatic gluconeogenesis, and increased FOXO1 and AMPK phosphorylation.	(Kim et al., 2020)	(Yang, Jang & Hwang, 2012)	
Anemarrhena asphodeloides	Glycosides	Mice	Inhibition of hepatic gluconeogenesis/glycogenolysis.	(Nakashima et al., 1993)	(Nurcahyanti et al., 2021)	
Annona stenophylla	Aqueous root extract.	Rats	Decreased glucose level.	(Taderera, Gomo & Shoriwa Chagonda, 2016)	(Taderera et al., 2019)	
Apios americana	Flower or methanolic extract of the flower.	Mice	Decreased plasma glucose level.	(Kawamura et al., 2015)	(Yan et al., 2017)	
Aronia melanocarpa	Fruit juice.	Rats	Decreased plasma glucose and triglycerides in diabetic rats.	(Lee et al., 2016; Mazibuko et al., 2013; Mu et al., 2020)	(Schreck & Melzig, 2021)	
Artemisia dracunculus	Ethanolic extract.	Mice	Lowered glucose and PEPCK concentrations.	(Ribnicky et al., 2006)		
Aspalathus linearis	Tea extract.	Mice	Improved impaired glucose tolerance.	(Kawano et al., 2009)	(Mazibuko et al., 2013)	
Boehmeria nivea	Methanol extract of the root.	Wistar rats	Restore normal glucose, lipids, and antioxidants level.	(Sancheti et al., 2011)	(Kim et al., 2013)	
Brassica oleracea	Raw sprouts.	Rats	Decreased blood glucose, glycated hemoglobin, and hepatoprotection.	(Sahai & Kumar, 2020)	(Schreck & Melzig, 2021)	
Cimicifuga racemosa	Rhizomes and root extract.	Mice	Reduced body weight, plasma, glucose, and increased insulin sensitivity.	(Moser et al., 2014)	(Moser et al., 2014)	
Cinnamomum cassia	Bark extract.	Diabetic mice	Decreased blood glucose and triglycerides levels.	(Kim, Hyun & Choung, 2006)	(Lakshmi et al., 2009)	
Citrullus colocynthis	Fruit ethanol extract.	Albino rats	Reduced blood glucose and improved pathology.	(Oryan et al., 2014)	(Drissi et al., 2021)	
Costus igneus (insulin plant)	Powdered leaves.	Rats	Decreased fasting and postprandial glucose level	(Shetty et al., 2010)	(Kattaru et al., 2021)	
Crataegus pinnatifida	Fruit extract.	Mice	Decreased glucose production and triglyceride synthesis via AMPK phosphorylation.	(Shih et al., 2013)	(Schreck & Melzig, 2021)	
Crocus sativus	Hydroethanolic extract of aerial parts.	Rats	Reduced blood glucose and improved diabetic complications.	(Ouahhoud et al., 2019)	(Kang et al., 2012)	
Curcuma longa	Curcuminoids and sesquiterpenoids from rhizome solvent fractions.	Mice	Decreased blood glucose levels and stimulation of adipocyte differentiation.	(Nishiyama et al., 2005)	(Kim et al., 2010)	
Dendrobium officinale	Stem extract.	Rats	Reduced blood glucose, total cholesterol, triglycerides, and LDLP-C.	(Chen et al., 2020)	(Wang et al., 2018)	
Entada phaseoloides	Entagenic acid from seed kernel.	Mice	Improved blood glucose, insulin resistance, and changes in pancreatic islets.	(Xiong et al., 2018)	(Zheng et al., 2016)	
Eucommia ulmoides	Leaves	Rats and Mice	Hypoglycemia and hypolipidemic effects in streptozotocin-induced hyperglycemia.	(Nakashima et al., 1993; Taderera, Gomo & Shoriwa Chagonda, 2016; Park et al., 2006; Lee et al., 2005)	(Schreck & Melzig, 2021)	
Gundelia tournefortii	Water extract.	Mice	Decreased blood glucose level, body weight, triglycerides, and cholesterol, but increased renal protection.	(Sancheti et al., 2011; Sahai & Kumar, 2020; Mohammadi & Zangeneh, 2018; Azeez & Kheder, 2012)	(Kadan et al., 2018)	
Juglans regia	Leaves and ridges.	Mice
Rats	Decreased blood glucose, hepatic phosphoenolpyruvate carboxykinase, glycogen phosphorylase activity, glycosylated hemoglobin, LDL, triglycerides, and total cholesterol.	(Kamyab et al., 2010; Liu et al., 2015; Sato et al., 2016)	(Schreck & Melzig, 2021)	
Juniperus chinensis	Berries ethanol extract.	Rats	Improved blood glucose level and other diabetic parameters.	(Ju et al., 2008)	(Jung et al., 2017)	
Kigelia pinnata	Methanolic extract of the flower.	Rats	Decreased blood glucose, serum cholesterol, and triglycerides.	(Kumar, Kumar & Prakash, 2012)	(Faheem et al., 2012)	
Malva verticulata	Tea	Mice	Decreased blood glucose, LDL-C, and total cholesterol and increased HDL-C and leptin.	(Bano & Akhter, 2021)	(Jeong & Song, 2011)	
Mangifera indica	Aqueous extract of the leaves.	Rats	Decreased fasting blood glucose level.	(Madhuri & Mohanvelu, 2017)	(Nandabalan, Sujatha & Shanmuganathan, 2010)	
Momordica charantia	Aqueous seed extract.	Rats	Reduced blood glucose, glycosylated hemoglobin, lactate dehydrogenase, glucose-6-phosphatase, fructose-1,6-biphosphatase, and glycogen phosphorylase, but increases the activities of glycogen synthase and hexokinase.	(Sathishsekar & Subramanian, 2005)	(Kumar et al., 2009)	
Momordica charantia	Saponins	Rats	Decreased fasting blood glucose, triglycerides, total cholesterol, and increased insulin content and sensitivity.	(Jiang et al., 2020)	(Cheng et al., 2008)	
Morus alba	Polysaccharides from fruit.	Rats	Reduced blood glucose and lipid levels.	(Jiao et al., 2017)	(Naowaboot et al., 2012)	
Ocimum basilicum	Aerial parts.	Rats	Inhibition of glycogenolysis.	(Ezeani et al., 2017)	(Kadan et al., 2016)	
Opuntia ficus-indica	Powder or water extract of the stem.	Rats	It inhibits α-glucosidase and reduces blood glucose levels.	(Hwang, Kang & Lim, 2017)	(Leem et al., 2016)	
Panax ginseng	Ethanol extract of the seed.	Obese diabetic mice	Increased insulin-stimulated glucose disposal, energy expenditure, and reduced cholesterol levels.	(Attele et al., 2002; Shalaby & Hammouda, 2013)	(Kang et al., 2017)	
Peumus boldus	Boldine alkaloid from the leaves and bark.	Rats	Dose-dependent decrease in oxidative markers and mitochondrial protection	(Jang et al., 2000)	(Schreck & Melzig, 2021)	
Portulaca oleracea	Aqueous extract.	Male Wistar rats	Decreased Hb A1C, serum glucose level, TNF-α, and IL-6.	(Ramadan, Schaalan & Tolba, 2017)	(Stadlbauer et al., 2016)	
Psidium guajava	Leaf extract.	Rats	Antidiabetic	(Mazumdar, Akter & Talukder, 2015)	(Li et al., 2019b)	
Punica granatum	Fruit aqueous extract.	Wistar rats	Reduces fasting blood glucose and lipid levels.	(Gharib & Kouhsari, 2019)	(Huang et al., 2005)	
Rhodiola crenulata	Methanol root extract.	Mice	Decreased postprandial blood glucose.	(Yue et al., 2022)	(Lee et al., 2015)	
Rosmarinus officinalis	Water extract.	Rats	Decreased blood sugar level and oxidative stress markers.	(Khalil et al., 2012)	(Vlavcheski et al., 2018)	
Salacia oblonga	Water extract of the root.	Obese Zucker rats	Improved interstitial and perivascular fibrosis and inhibition of postprandial hyperglycemia.	(Li et al., 2004)	(Giro et al., 2009)	
Sapindus mukorossi	Fruit	Rats	Decreased glucose and lipid levels.	(Verma et al., 2012)	(Stadlbauer et al., 2021)	
Sarcopoterium spinosum	Aqueous extract.	Mice	Prevents diabetes progression.	(Smirin et al., 2010)	(Elyasiyan et al., 2017)	
Sechium edule	Methanol and ethyl acetate fraction.	Rats	Antidiabetic and antioxidant.	(Siahaan et al., 2020)	(Wu et al., 2014)	
Selaginella tamariscina	Total flavonoids	Rats	Decreased plasma FBG, HbA1c, triglycerides, total cholesterol, FFA with increased insulin, HDL-C, and C-peptides.	(Zheng et al., 2011)	(Nguyen et al., 2015b)	
Stauntonia chinensis	Total saponins from the stem.	Mice	Hypoglycemic and hypolipidemic.	(Xu et al., 2018)	(Hu et al., 2014)	
Toona sinensis	Quercetin from the leaves.	Mice	Antidiabetic and antioxidant.	(Zhang et al., 2016)	(Liu et al., 2015)	
Trigonella foenum-graecum	Seed powder.	Female Albino rats	Reduced elevated fasting blood glucose and enzyme levels.	(Raju et al., 2001)	(Chen et al., 2022)	
Urtica dioica	Aqueous extract of the aerial parts.	Wistar rats and Swiss mice	Decreased glucose level in oral glucose tolerant test [OGTT].	(Bnouham et al., 2003)	(Chen et al., 2022)	
Vaccinium myrtillus	Fruit	Rats	Decreased total cholesterol, LDL-C, VLDL-C, and triglycerides in alloxan-induced hyperglycemic rats.	(Asgary et al., 2016)	(Schreck & Melzig, 2021)	
Vigna angularis	Hot water extract and polysaccharides from the leaves.	Mice and Rats	Reduced FBG, an triglycerides, but increased HDL-C, and reduction in diabetes progression.	(Zheng et al., 2011; Xu et al., 2018; Itoh et al., 2009)	(Sato et al., 2016)	
Zea mays (Purple corn)	Extract	Mice	Decreased fasting blood glucose, HbA1c, and PEPCK, increased insulin secretion, AMPK and GLUT4 in diabetic mice.	(Huang et al., 2015)	(Luna-Vital & De Mejia, 2018)	
Gymnema sylvestre	Phytoconstituents	Rats	Reduced hyperglycemia via through PI3K/AKT activation	(Li et al., 2019a)	(Retz & Glucose, 2021)	

Aronia melanocarpa fruit juice was found to mediate a dose-dependent decrease in plasma glucose and triglyceride levels in streptozotocin-induced hyperglycemic rats (Lee et al., 2016; Mazibuko et al., 2013), corresponding to the observations on glucose transport in Caco-2 cells alluded to above (Table 1) (Schreck & Melzig, 2021). Similarly, the antidiabetic and hyperlipidemic effects of Crataegus pinnatifida were investigated in high fat-fed mice. The results showed decreased glucose production and triglyceride synthesis via induction of AMPK phosphorylation (Shih et al., 2013), compared with inhibition of SGLT1 and GLUT2 in Caco-2 cells (Schreck & Melzig, 2021). These provide a good example of studies in cell lines being translated into animal models.

The fruit of Vaccinium myrtillus was reported to significantly reduce serum glucose, total cholesterol, low density lipoprotein cholesterol, and very low density lipoprotein cholesterol, and triglycerides in alloxan-induced hyperglycemic adult male Wistar rats (Asgary et al., 2016).

Juglans regia extracts reduce blood glucose levels in diabetic mice (Kamyab et al., 2010), ameliorated streptozotocin-induced diabetic peripheral neuropathy in rats (Nasiry et al., 2017), and significantly decreased blood glucose, glycosylated hemoglobin, LDL, triglycerides, and total cholesterol in Wistar rats (Mohammadi et al., 2011). An aqueous extract of the seeds of Momordica charantia reduced blood glucose level, glycosylated hemoglobin, lactate dehydrogenase, glucose-6-phosphatase, fructose-1,6-biphosphatase, and glycogen phosphorylase but increased the activities of glycogen synthase and hexokinase in streptozotocin-induced diabetic rats, providing clear evidence of a systematic and programmed action on key metabolic activities (Sathishsekar & Subramanian, 2005). Similarly, polysaccharides of Dendrobium officinale reduced blood glucose level, glycated serum protein, total cholesterol, LDL-C, and increased HDL-C in type 2 diabetic rats (Chen et al., 2020). Cimicifuga racemosa extracts from rhizomes and roots reduced body weight, plasma glucose, improved glucose metabolism, and increased insulin sensitivity in obese diabetic mice (Moser et al., 2014). Tarralin™, an ethanolic extract of Artemisia dracunculus, significantly lowered blood glucose concentrations and PEPCK in diabetic KK-Ay mice (Ribnicky et al., 2006). Li et al. (2019a) described the effects of Gymnemic acid isolated from G. sylvestre on insulin signalling pathways in the type 2 diabetic rats as activation of PI3K/AKT together with AMPK phosphorylation. Such studies exemplify the power of medicinal plants in the amelioration of metabolic disturbances, and Table 5 summarizes the wide array of studies relevant to diabetes research.

Medicinal plants in clinical trials

The process of drug discovery necessitates that a drug molecule or product that has successively passed through the preclinical stage of drug development is carefully tested in clinical trials. We reviewed those plants that progressed to clinical trials and present our findings in Table 7. As this area is particularly important, we have provided some detail of key studies in the sections below.

Table 7 Medicinal plant antidiabetics from cell-biology to clinical trial.

Studies in which the indicated plants were examined in clinical trials.

Medicinal plants	Phytochemistry
Product	Clinical trial	References	
Aronia melanocarpa	Alixir 400 PROTECT®
[Standardized extract]	Prospective open-label trial of 148 patients.	(Tasic et al., 2021)	
Cinnamomum cassia	Extract and 1,000 mg capsule.	Randomized-placebo control of 70 patients and another study of 19 subjects.	(Hasanzade et al., 2013, Mustafa et al., 2017)	
Citrullus colocynthis	Fruit capsule.	Randomized clinical trial of 50 T2D patients.	(Jang et al., 2008)	
Crataegus pinnatifida	Multi-herb	Randomized double-blind, placebo-controlled trial of 40 patients.	(Hu, Zeng & Tomlinson, 2014)	
Curcuma longa	500 mg/day	Randomized double blind, placebo-controlled trial of 71 patients	(Neta et al., 2021)	
Gundelia tournefortii	250 mg of hydroalcoholic extract of the aerial parts.	Randomized double-blind, placebo-controlled trial of 38 patients.	(Hajizadeh-Sharafabad et al., 2016)	
Juglans regia	Leaf extract 100 mg twice daily and a hydroalcoholic leaf extract.	Randomized double-blind, placebo-controlled trial of 61 and 50 patients.	(Taghizadeh et al., 2022; López-Romero et al., 2014)	
Mangifera indica	Low dose [0.5 g/kg] and high dose [1 g/kg] of the leaf extract.	Clinical investigation of 26 T2DM patients.	(Waheed, Miana & Ahmad, 2006)	
Momordica charantia	2,000 mg/day of dried powder of fruit.	Randomized, double-blind, placebo-controlled trial of 24 patients.	(Cortez-Navarrete et al., 2018)	
Morus alba	300 mg extract.	Randomized clinical trial of 60 type 2 diabetic patient [T2DM].	(Taghizadeh et al., 2022)	
Ocimum basilicum	Raw and processed seeds.	45 days clinical trial using convenient sampling.	(Arivuchudar, Nazni & Uvaraj, 2022)	
Opuntia ficus-indica	Nopal [Opuntia ficus-indica preparation].	Clinical study.	(López-Romero et al., 2014)	
Panax ginseng	Extract of fermented root and Korean red ginseng preparation.	Randomized clinical trial of 42 subjects and double-blind randomized crossover design of 19 subjects.	(Vuksan et al., 2008)	
Portulaca oleracea	Seeds	Clinical study of 30 patients and randomized trial of 74 subjects.	(El-Sayed, 2011; Darvish Damavandi et al., 2021)	
Punica granatum	Dried flower mouth wash.	Randomized trial of 80 diabetes patients with gingivitis.	(Sedigh-Rahimabadi et al., 2017)	
Salacia oblonga	Extract [240, 480 mg/kg]	Randomized double-blind crossover trial of 60 patients.	(Williams et al., 2007)	
Trigonella foenum-graecum	Seed capsule.	Multicenter randomized, placebo-controlled, double-blind, add-on clinical trial of 154 T2D patients. Another 12 weeks trial of 12 patients.	(Verma et al., 2016; Najdi et al., 2019)	
Urtica dioica	Ethanolic extract.	Double-blind, randomized trial of 50 diabetic women.	(Amiri Behzadi, Kalalian-Moghaddam & Ahmadi, 2016)	
Gymnema sylvestre	Leaf water extract	22 non-insulin dependent diabetic patients	(Baskaran et al., 1990)	

Prospective open-label clinical trials of Alixir 400 PROTECT® (standardized extract of Aronia melanocarpa) in 143 patients demonstrated controlled glycemia, blood pressure improvement, and beneficial effects on LDL-C, triglycerides and total cholesterol, and was of significant (p < 0.05) overall benefit in diabetic hypertensive patients (Tasic et al., 2021). Similarly, a meta-analysis of controlled clinical trials carried out on Aronia melanocarpa daily supplementation revealed significant (p < 0.05) decreases in total cholesterol, blood pressure and a reduction in cardiovascular and diabetic risk factors, clearly supporting a useful role in therapy (Hawkins et al., 2021).

Lipid lowering effects are also a common feature of clinical trials with medicinal plants. Hu, Zeng & Tomlinson (2014) demonstrated the beneficial effects of a multi-herb formula containing Crataegus pinnatifida for dyslipidemia in a randomized double-blind, placebo-controlled trial, reporting decreased plasma lipids, glucose levels, HbA1c, and LDL-C at 95% CI. A randomized, double-blind placebo-controlled trial of Juglans regia leaf extract resulted in a significant decrease (p < 0.05) in fasting blood glucose levels, triglycerides, total cholesterol and HbA1c compared with placebo (Hosseini et al., 2014). Similar beneficial effects were reported in further trials (Rabiei et al., 2018) including beneficial effects in patients with coronary artery disease with significant decreases (p = 0.04) in total cholesterol, BMI, and LDL (Hajizadeh-Sharafabad et al., 2016). A clinical investigation of oral administration of Portulaca oleracea seeds in 30 T2DM subjects revealed a significant decrease (p < 0.001) in serum levels of triglycerides, total cholesterol, LDL-C, and liver enzymes, but increased (p > 0.001) levels of HDL-C and albumin (El-Sayed, 2011). Purslane (Portulaca oleracea capsule) also drove a significant difference (p > 0.01) in the triglycerides, liver enzymesand fasting blood glucose in 74 people with T2DM in a randomized double-blind, placebo-controlled clinical trial. An improvement in both insulin resistance, and LDL-C levels was also reported (Darvish Damavandi et al., 2021).

Decreased blood glucose levels are often used as a key outcome. A 14-day clinical investigation involving 26 people with T2DM on a low (0.5 g/kg) and high (1 g/kg) doses of aqueous and alcoholic extract of the powdered leaves of Mangifera indica showed a significant decrease in blood glucose levels in all groups (Waheed, Miana & Ahmad, 2006). Similarly, both raw and processed seeds of Ocimum basilicum in patients with diabetes and dyslipidemia revealed beneficial effects including decreased blood glucose, a reduction in body mass index, triglycerides, LDL-C, and decreased HDL-C at 5% and 1% levels of significant (Arivuchudar, Nazni & Uvaraj, 2022). A significant decrease (p < 0.05) in the fasting blood glucose and HbA1c levels were observed in people with T2DM after 2 month treatment with the fruit capsule of Citrullus colocynthis (Jang et al., 2008).

In a randomized, double-blind, crossover study of 60 diabetic subjects receiving Salacia oblonga extract, Williams et al. (2007) reported significant decrease (p < 0.05) in glyceamia and insulinemia in patients after high carbohydrate meal. A 4-week randomized double-blind, placebo-controlled clinical trial of fermented red ginseng (Panax ginseng) involving 42 patients with impaired fasting glucose or T2DM also showed significant decrease (p < 0.01) in postprandial glucose levels and increased postprandial insulin levels compared to the placebo group (Oh et al., 2014). A further study supported these conclusions (Vuksan et al., 2008).

Many other studies, highlighted in Table 7, have shown pronounced and beneficial effects.

Not all data are conclusive. The results of a 60-day randomized-placebo clinical trial by Hasanzade and colleagues using Cinnamomum cassia in 70 people with T2D revealed no significant difference (p > 0.05) between the test and placebo (Hasanzade et al., 2013). On the other hand, Hoehn and Stockert in a smaller trial reported significant decrease in blood sugar levels of the patients taking 1,000 mg Cinnamomum cassia capsule for 12 weeks (Hoehn & Stockert, 2012). An aqueous extract of G. sylvestre (GS4) 400 mg/day used over 18 to 20 months supplementation drove significant decreases (p < 0.001) in blood glucose, glycosylated haemoglobin and glycosylated plasma protein in 22 patients (Baskaran et al., 1990). It should also be clearly noted that many of the clinical studies performed to date include relatively small numbers of patients. Larger studies will provide impetus for more work in this area.

Conclusion and future perspectives

Diabetes mellitus-related morbidity and mortality continues to increase globally and necessitates urgent action to identify and drive novel therapies which can be widely used in under-developed economies. Our review reveals that antidiabetic drugs of herbal origin can play a modulatory role in insulin signaling pathways and drive metabolically relevant changes in insulin action, such as elevated glucose transport. Tissue culture systems have provided key insight into the molecular mechanisms of the phytochemicals beneficial to diabetic patients and have contributed both mechanistic insight and facilitated the development of more clinically-facing treatments. Among the plants we reviewed in tissue culture systems, close to half (45%) have been investigated for their antidiabetic activities in mammals (rats, mice, and rabbits) and 4% have been tested in human clinical trials. The positive outcomes reported in these clinical trials should be recognized as providing a new impetus to phytobiology research as an effective treatment for insulin resistance and diabetes. In future, larger-scale clinical trials are clearly warranted given the largely positive effects of many of these natural products. There is a need to screen larger numbers and citizens of different genetic backgrounds to identify potential population-specific benefits. Similarly, the coupling of phytochemical studies to genomic data may offer a powerful means to develop combination therapies and more personalized medicine approaches.

List of Abbreviations

ACC Acetyl-CoA carboxylase

Akt Protein kinase B

AMPK Adenosine monophosphate-activated protein kinase

AS160 Akt substrate 160 kDa

ATPase Adenosine triphosphatase

CI Confidence interval

GCK Glucokinase

GLP-1 Glucagon-like peptides-1

GLUT Glucose transporter

GSK-3 Glycogen synthase kinase 3

GSV Glucose storage vesicle

HbA1c Hemoglobin A1c

HDL High-density lipoprotein

HSC Hematopoietic stem cell

IFN Interferon

IR Insulin receptor

IRS Insulin receptor substrate

LDL Low-density lipoprotein

LPS Lipopolysaccharide

PEPCK Phosphoenolpyruvate carboxykinase

PI3K Phosphoinositide-3 kinase

PPARγ Peroxisome proliferator-activated receptor gamma

PTP1B Protein tyrosine phosphate 1B

SGLT1 Sodium-glucose linked transporter1

T1DM Type 1 diabetes mellitus

T2DM Type 2 diabetes mellitus

Additional Information and Declarations

Competing Interests

Author Contributions

Data Availability

Gwyn W. Gould is an Academic Editor for PeerJ.

Simeon Omale conceived and designed the experiments, performed the experiments, analyzed the data, prepared figures and/or tables, authored or reviewed drafts of the article, and approved the final draft.

Kennedy I. Amagon conceived and designed the experiments, performed the experiments, analyzed the data, authored or reviewed drafts of the article, and approved the final draft.

Titilayo O. Johnson conceived and designed the experiments, performed the experiments, analyzed the data, authored or reviewed drafts of the article, and approved the final draft.

Shaun Kennedy Bremner analyzed the data, authored or reviewed drafts of the article, and approved the final draft.

Gwyn W. Gould conceived and designed the experiments, prepared figures and/or tables, authored or reviewed drafts of the article, and approved the final draft.

The following information was supplied regarding data availability:

This is a literature review.

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
