# Peer review of "A systematic analysis of anti-diabetic medicinal plants from cells to clinical trials"

_PeerJ, doi:10.7717/peerj.14639_

## Round 0.1 · original submission · Minor Revisions

Please revise the manuscript as per the reviewers' comments.

Reviewer 2 has suggested that you cite specific references. You are welcome to add it/them if you believe they are relevant. However, you are not required to include these citations, and if you do not include them, this will not influence my decision.

Reviewer 1 ·

Basic reporting

1. Searches were only performed between December 2021 and April 2022. What’s the consideration of restricting search in this time frame? Is this too short? Please clarify.

2. The authors excluded papers not freely available via our institution (in this case the University of Strathclyde). How many papers did the author exclude due to this reason?

3. The search terms are broad, but it’s not clear about the outcomes of interest in this study. What are the specific outcomes/endpoints the authors searched? Please clarify and elaborate.

4. Appreciate the authors provide a detailed review of the use of medicinal plants in clinical trials. But the review is more descriptive and lack of quantitative evidence. For example, in line 450-455, the authors wrote “…demonstrated controlled glycemia, blood pressure improvement, and beneficial effects on LDL-C, …, and was of significant overall benefit in diabetic hypertensive patients. Similarly, a meta-analysis of controlled clinical trials … revealed decreased total cholesterol, blood pressure and a reduction in cardiovascular and diabetic risk factors, clearly supporting a useful role in therapy”. It’s not clear what the effect size is, if significant what the p-value and confidence interval is and what the context/reference for effects are, which makes readers hard to interpret the results.

5. The randomized trials the authors reviewed in the manuscript only include a relatively small number of patients (e.g., less than 100 patients), which is much smaller than the sample size in the typical randomized trials for commonly used treatments for diabetes. Thus, the evidence summarized in the manuscript, in my view, are limited; however, the authors did not discuss the limitations of these studies sufficiently.

Experimental design

See comments above

Validity of the findings

See comments above

Reviewer 2 ·

Basic reporting

1. Authors may discuss well-researched medicinal plants such as Gymnema sylvestre and glycyrrhiza uralensis, as they are more thoroughly studied from the perspective of diabetes.
2. Including a section on the “future perspective” will increase reader’s interest.
3. Authors are suggested to discuss the skeletal muscle perspective in more details as it is the
Most important metabolic tissue accountable for the majority of insulin-mediated glucose uptake. Here are few reference papers for example,
https://doi.org/10.2174/1389203720666191119100759, https://doi.org/10.1177/2515690X21100633, https://doi.org/10.3390/ijms21113845, https://doi.org/10.1080/87559129.2022.2087669

4. The conclusion should be re - written to make it more particular and impactful.
5. The sentence “Our review on cell-biology based research on the different molecular mechanisms of antidiabetic drugs of herbal origin revealed that phytochemicals play a modulatory role in both effector and insulin signaling pathways.” is not appropriate and meaningful.

Experimental design

This is an thorough and well-written review.

Validity of the findings

NA

·

Basic reporting

The manuscript is written in clear and unambiguous professional English. Manuscript include sufficient introduction and background which in clear and coherent way demonstrate the knowledge in the field of T2DM treatment, both by the traditional and plants origin drugs. The introduction make the subject clear. Relevant prior literature was referenced appropriately. Figures and tables are relevant to the content of the manuscript and support the understanding of the described research. What’s more tables and figures are appropriately described and labeled.
The manuscript is broad interest, because authors reviewed both in vitro and in vivo studies regarding the use of plant origin extracts with antidiabetic properties and also the drugs based od the plants used in clinical trials.

Experimental design

The manuscript is a systematic review. In the section Methodology authors described the way of data searching by the use several of scientific database. It is lack of the flow diagram demonstrated how the selection procedure of the literature was made. Apart from that, the manuscript is organized logically into coherent subsections.

Validity of the findings

The great advantage of reviewed manuscript is That the authors collected data on the antidiabetic properties of medicinal plants , ranging from cell studies to animal studies and clinical studies.

The conclusions was appropriately stated, it is connected to the original question investigated, and is limited to those supported by the results.

---

## Round 0.2 · accepted · Accept

In the revised manuscript authors addressed all of the reviewers' comments. After reviewing all the comments and overall evaluation, the manuscript found satisfactory. Now. it may be accepted for publication.